# Effects of 'The Vicious Worm' educational software on *Taenia solium* knowledge among key pork supply chain workers in Zambia

**Victor Vaernewyck**[1], **Kabemba Evans Mwape**[2], **Chishimba Mubanga**[1,2], **Brecht Devleesschauwer**[1,3], **Sarah Gabriël**[1☯], **Chiara Trevisan**[4☯]*

**1** Department of Veterinary Public health and Food Safety, Faculty of Veterinary Medicine, Ghent University, Ghent, Belgium, **2** Department of Clinical Studies, School of Veterinary Medicine, University of Zambia, Lusaka, Zambia, **3** Department of Epidemiology and Public Health, Sciensano, Brussels, Belgium, **4** Department of Biomedical Sciences, Institute of Tropical Medicine, Antwerp, Belgium

☯ These authors contributed equally to this work.
* ctrevisan@itg.be

**Data Availability Statement:** All relevant data are within the manuscript and its Supporting Information files.

## Abstract

The neglected zoonotic cestode *Taenia solium* is endemic in many low- and middle-income countries, including Zambia. The parasite infects humans and pigs, inflicting high socioeconomic and disease burdens in endemic areas. Health education is regarded as an important component in *T. solium* control and previous studies indicate that 'The Vicious Worm' may be an effective *T. solium* health education tool for Tanzanian medical and agricultural professionals and Zambian primary school students. This study aimed to assess the effects of health education using 'The Vicious Worm' among Zambian pork supply chain workers, because the pork supply chain greatly influences food safety and security in Zambia. Half-day educational workshops using 'The Vicious Worm' and subsequent follow-up sessions were organized in the Lusaka and Katete districts of Zambia in March and April 2019. Questionnaires were administered before, after, and three weeks after the use of 'The Vicious Worm' to assess the program's impact on knowledge uptake and short-term retention. Focus group discussions were conducted to assess the program's user experience and the participants' beliefs, attitudes, and insights. In total, 47 pork supply chain workers participated: 25 from Lusaka and 22 from Katete. Overall, knowledge about *T. solium* was significantly higher (p<0.001) both immediately after, and three weeks after the use of 'The Vicious Worm' compared to baseline knowledge. Focus group discussions indicated incipient attitudinal and behavioral change, as well as a positive reception of the software; with participants describing the software as simple, educative, and useful to share knowledge. The study results indicate that workshops using 'The Vicious Worm' may be effective for short-term *T. solium* health education among key pork supply chain workers. Follow-up studies are required to assess long-term effects, transfer of knowledge and behavioral change. However, educational interventions with 'The Vicious Worm' could be considered for integrated *T. solium* control programs in sub-Saharan Africa, especially if the educational content is further simplified and clarified.

**Funding:** The research was part of the CYSTISTOP project made possible with financial support from 'ITM's SOFI programme supported by the Flemish Government, Science & Innovation' (https://www.ewi-vlaanderen.be/sites/default/files/evaluation_2018-2019-research_assignment_itm-english-summary.pdf) grant number 123456. VV was supported by the Faculty Mobility Fund (https://www.ugent.be/di/en/research/mobility/overview.htm) grant 2018-S-03. The funders had no role in study design, data collection and analysis, decision to publish, or preparation of the manuscript.

**Competing interests:** The authors have declared that no competing interests exist.

## Author summary

The tapeworm *Taenia solium* causes epilepsy and severe headaches in humans and economic losses to smallholder farmers in endemic areas where free-roaming pigs, poor sanitation and informal animal slaughter are prevalent. Treatment of human taeniosis and interventions in pigs (vaccination and anthelmintic treatment) have been established as essential tools to achieve short-term control, but health education will be crucial to sustain long-term control. 'The Vicious Worm' is a specific *T. solium* health education tool aiming to convey simple and meaningful messages concerning disease prevention and control. Previous studies found significant knowledge increases after the use of the educational tool in Tanzanian medical and agricultural professionals, and Zambian primary school students. We organized half-day educational workshops using 'The Vicious Worm' for pork supply chain workers at two study sites in Zambia, with follow-up visits three weeks later. We found a significant increase in participants' knowledge, both immediately and three weeks after the health education. Furthermore, the program was well received and potentially led to attitudinal and behavioral change that could deter the propagation of *T. solium*. We conclude that 'The Vicious Worm' educational interventions may contribute to a safer pork supply chain and we encourage its implementation in future *T. solium* control strategies.

## Introduction

*Taenia solium*, the pork tapeworm, is a parasite that imposes a substantial burden on the health and livelihoods of subsistence farming communities in developing countries of Asia, Latin America and sub-Saharan Africa, including Zambia [1]. This cestode is the causative agent of *T. solium* taeniosis/cysticercosis (TSTC), a neglected zoonotic disease complex that accounts for the highest global burden of foodborne parasitic diseases and acts as a leading cause of deaths from foodborne diseases [2–4]. It is a significant contributor to late-onset epilepsy in tropical regions around the world [5, 6], and causes pig production losses, driving smallholder farmers into poverty or even destitution [7, 8]. The importance of *T. solium* has been recognized by the World Health Organization, which listed TSTC as a major neglected tropical disease (NTD) and has targeted it for control [9].

The human definitive host acquires the intestinal tapeworm infection (taeniosis) by ingestion of raw or undercooked pork containing viable cysticerci. The adult worm produces infective eggs that are passed in the host's stool, contaminating the environment. Pigs act as intermediate hosts after ingestion of these eggs and develop porcine cysticercosis (PCC), a tissue infection with larval cysts (cysticerci). Humans can act as dead-end intermediate host (human cysticercosis, HCC); with larval cysts commonly found in the central nervous system (neurocysticercosis, NCC) [1].

Several studies in Zambia reported high prevalences of PCC, HCC and taeniosis, confirming a high *T. solium* endemicity [10–14]. PCC antigen prevalences of 28.3%, 16.9% and 30.0% were found for the Southern, Eastern and Western Province of Zambia respectively when 1,691 pigs were examined [10]. Like in many other developing countries, there is an increasing demand for pork in Zambia, especially in rural areas [15, 16] where pigs are mostly kept under smallholder conditions as a low input transitory activity with pigs roaming freely [17]. Moreover, pig slaughter often occurs informally in backyards or at illegal slaughter establishments with little or no controlled meat inspection [15]. This pork supply chain creates a favorable

environment for the propagation of *T. solium*, which is further aggravated by a reluctance to build and use latrines by most Zambian rural households [18]. Addressing practices at the level of pork production and distribution may prove valuable *T. solium* control points. Indeed, previous PRECEDE-PROCEDE [19] assessments have identified pork producers and sellers as important target groups for *T. solium* health promotion strategies [20, 21].

Lack of knowledge is an acknowledged barrier for control. Previous *T. solium* health education interventions in India [22], Kenya [23], Mexico [24], Tanzania [20, 25–29] and Zambia [30, 31] led to significant increases in knowledge among targeted stakeholders. Moreover, attitudinal and behavioral change were reported in different *T. solium* health education studies; including increased condemnation of infected meat, better confinement of pigs and improved sanitary/hygienic practices [20, 22–25, 27–30, 32]. More importantly, health education proved able to reduce PCC [20, 24, 25] and HCC [32] incidence. An additional advantage of health education is that the promotion of good sanitation and hygiene also decreases the disease burden of many other sanitation-related pathogens [1, 33, 34], such as soil transmitted helminths [35] and diarrheal agents [36].

Different health education approaches have been tested; including the use of informational movies, street plays, songs, interactive discussions, leaflets, posters, banners, trained teachers and software programs [22–28, 30, 31]. 'The Vicious Worm' (TVW, https://theviciousworm. be/) is a free educational software program that provides evidence-based information on *T. solium* through illustrated short stories, videos, quizzes and scientific texts. It aims to raise awareness of *T. solium* and reduce risk behaviors such as open defecation, keeping pigs free-ranging and cooking pork inadequately. The program supports standardized education while three dedicated levels (village, town and city level) allow more targeted conveyance of information to different stakeholders (laypeople, professionals and policy makers) [37]. The software is currently available as computer program and mobile application in English and Swahili (Minyoo Matata) [37, 38].

Previous studies found that health education with TVW provided significant short- and long-term knowledge uptake in Zambian primary school students [30, 31] and Tanzanian medical and agricultural professionals [26, 28]. Furthermore, TVW was well-received by study participants and exposure to the program was reported to lead to positive behavioral changes and effective knowledge dissemination [20, 26, 28, 30]. Our study aimed to assess the effects of TVW health education on key actors in the Zambian pork supply chain.

## Methods

### Ethics statement

Ethical clearance was obtained from the Excellence in Research Ethics and Science Converge (ERES) Institutional Review Board of Lusaka, Zambia (Approval No. 2018-Dec-012) and the Ethical Committee of the University of Antwerp, Belgium (Approval No. B300201628043, EC UZA16/8/73).

The study was explained to the participants at the start of the educational workshop, before names and phone numbers (if applicable) were registered on an ID list to facilitate future contact. Written informed consent was sought from the individual participants. For illiterate participants, a thumbprint was used and a witness signed on their behalf. Participation did not involve any noteworthy risk, and all collected data was processed confidentially.

Attendance to the educational sessions was voluntary and participants were free to leave the study at any time. The workshops took place during working hours; hence permission was sought from the employers. Each participant was offered a free beverage and snack during the sessions, and a small monetary incentive for participation and lost working time.

## Study area

The study was carried out at two study sites in Zambia, where the national pig production quadrupled between 2000 and 2018 (0.3 million to 1.2 million) [39].

The first study site was Lusaka district (Lusaka province), where pig slaughter and meat cutting often took place at an unofficial slaughter slab, before the pork was sold at the establishment or taken to markets in surrounding high density areas within the city. Most of these animals were village pigs from resource-poor farmers, largely originating from rural areas of the Southern Province of Zambia [11, 40, 41]. A previous study in 868 village pigs slaughtered at this slaughter slab, found a PCC prevalence of 64% [11].

The second study site was Katete district (Eastern province), where most households performed home slaughter [14, 42, 43]. A 2015 study in this district found a PCC prevalence of 46% (17/37) in randomly selected slaughter age pigs based on full carcass dissection, which is regarded as the golden standard detection technique [12]. Other studies in the district found very high prevalences of HCC/NCC [14, 44] and indicated NCC as the single most important cause of epilepsy in the study area [44].

## Study design

The cross-sectional study was conducted in March and April 2019. At both study sites, an educational workshop was organized with a subsequent follow-up session after three weeks.

The educational workshop included a brief study introduction; participant registration and signing of informed consent forms; a short survey on the participant's general information (7 questions); a 'pre' questionnaire survey; a short break; an educational component using TVW; a dialogue addressing questions about TVW; and a 'post' questionnaire survey. After the workshop, USB sticks with TVW software were handed out to participants with access to computers, facilitating the continuation of their training and knowledge sharing.

The follow-up session consisted of a '3weekspost' questionnaire survey and focus group discussions (FGDs). Additionally, observational data were collected at three time points per study site throughout the seven-week study period (Fig 1).

For the Lusaka study site, pork supply chain workers were gathered at the unofficial Chibolya slaughter slab, where the observational data were also collected. For the Katete study site, pork supply chain workers from the area were gathered at a local lodge, while observational data was collected throughout the district.

## Selection of study subjects

All professionals who were involved in the pork supply chain at the level of live trading, slaughtering, meat cutting and distribution in the study area were eligible to participate in the study. These selection criteria included professional pig traders, slaughterslab workers, butchers, and meat inspectors. Slaughterslab workers include the professionals who are responsible for the pig slaughter, the meat cutting, the selling of pork and the management at the slaughter slab facility. Professionals who reared pigs (e.g. pig farmers) and/or cooked pork (e.g. cooks), but

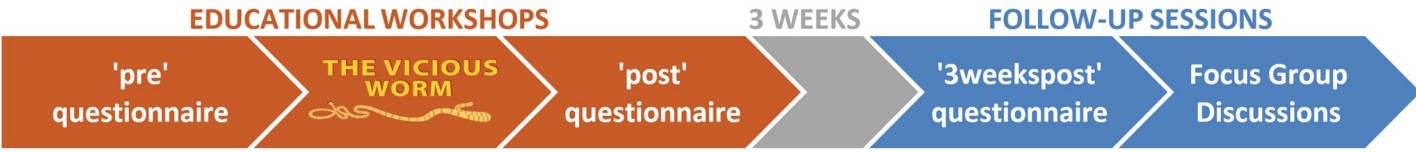

**Fig 1. Simplified study timeline.** Observational data were collected throughout the seven-week study period.

were not active in live trading, slaughtering, meat cutting or distribution, were excluded. Each participant was assigned an individual code, enabling confidential data processing.

## Questionnaires

The 'pre', 'post' and '3weekspost' questionnaires consisted of 22 multiple choice questions about *T. solium* (S1 File): 19 quantitative and 3 qualitative questions, based on the previously validated questionnaire [45]. The quantitative questions aimed to test the participants' knowledge about *T. solium* and were coded into six categories: *T. solium* in general, acquisition & transmission, PCC in live pigs, PCC in slaughtered pigs, taeniosis and HCC/NCC. Furthermore, the influence of participants' characteristics on their baseline knowledge and knowledge uptake was assessed using these quantitative questions in combination with the 7 questions concerning the participant's general information. There were two to six answer options per quantitative question with one correct answer. The qualitative questions were used to obtain information about the perceived PCC occurrence and the handling of PCC cases. During all three surveys, the same questions were asked, but the 'post' and '3weekspost' questionnaires had a different question sequence compared to the 'pre' questionnaire. The questionnaire was written in English and a monochrome paper copy was handed to each participant, in addition to a pen. The questionnaire was projected in color on a white cloth surface while the moderator read the questions aloud in English and translated them in the local language (Nyanja-Chewa). The questions were repeated until they were clear to the participants.

## Educational component using 'The Vicious Worm'

Approximately one hour and a half were assigned to the educational component of the study, with most attention devoted to TVW's 'village level', where scenes from an African village are depicted. In this level, information is presented in a simple manner, through illustrated short stories. Furthermore, the slaughterhouse section from the 'town level' was used to give more detailed and technical information about pig slaughter, meat inspection and good meat hygiene practices.

A fifteen-minute break with light refreshments was included between the pre-questionnaire and educational component. The English version of the software was projected on a white cloth as the moderator navigated through the program, reading (English) and translating (Nyanja-Chewa) the visible content.

## Focus group discussions

A total of four FGDs were conducted in the local language (Nyanja-Chewa) after the participants had completed the '3weekspost' questionnaire. Each group consisted of one moderator and nine to twelve study subjects. In the four FGDs, the participants' opinion of the program was evaluated, as well as their beliefs, attitudes, and insights concerning *T. solium*. Furthermore, participants were given the opportunity to suggest improvements and alternatives to the health education tool, and express the needs and wishes of local pork supply chain workers to enhance *T. solium* control. Furthermore, video footage of seizures in a human (3 min. fragment) and two pigs (1 min. and 2 min. fragments) was projected on a white cloth surface. Afterwards, participants were asked whether they considered it useful and acceptable to use such footage in TVW to highlight the severity of the disease and the link between HCC and PCC. All discussions were video recorded to facilitate the transcription of the discussions involving several individuals at the same time. A backup audio recording was made with a dictaphone app on a smartphone to ensure full coverage of the FGDs.

## Data management and statistics

The acquired quantitative and qualitative data were analyzed to assess knowledge uptake and short-term knowledge retention among key actors in the Zambian pork supply chain, and their attitude towards TVW. Furthermore, it inquired about possible improvements and alternatives to the software, the acceptability of using video footage demonstrating seizures in humans and pigs in TVW, and the needs and wishes of pork supply chain workers to enhance *T. solium* control.

## Questionnaires

The questionnaire data was entered in an Excel spreadsheet (Microsoft Office Professional 2016) where questions (q) were assessed individually and by category (c) (S1 Data). Each quantitative question was assigned a score of 1 when answered correctly, or 0 when answered incorrectly. When questions were not answered or when the indications were unclear, NA was assigned. For each questionnaire, data were independently entered twice in separate excel files. Both entries were subsequently compared using Spreadsheet Compare (Microsoft Office Professional 2016) and mismatches were revised. To simplify the comparison of the answers at different time points, each question was assigned a code from q0 to q22, based on their sequence in the pre-questionnaire. Similarly, the 19 quantitative questions were assigned a second code from c1 to c6, to estimate the knowledge per category at each time point.

To assess knowledge uptake between the 'pre', 'post', and '3weekspost' time points, we fitted generalized linear mixed models with logit link to the questionnaire results (number of correct answers over total number of completed answers). We used participant ID as a random effect to account for within-individual clustering. The period time point was included as a fixed effect. We used Tukey's all-pair comparisons to assess all possible pairwise differences between the three time points (i.e., post vs 'pre', '3weekspost' vs 'pre', and '3weekspost' vs 'post').

We also assessed the effect of covariates on baseline knowledge and on knowledge uptake ('post' vs 'pre'). For the effect on baseline knowledge, we fitted similar models as above, but now using the specific covariate as fixed effect, and limiting the dataset to the pre responses only. For the effect on knowledge uptake, we fitted the above-mentioned models with addition of an interaction effect between period time point and the specific covariate. For all models, we reported odds ratios (OR), and corresponding 95% confidence intervals (CI) and p-values.

All analyses were performed in R [46], using the 'lme4' [47] and 'multcomp' [48] packages.

## Focus group discussions

The recordings were transcribed and translated into English and each focus group was assigned a code for data analysis. Data analysis was performed using NVivo qualitative data analysis software [49]. All data was coded (172 codes with 290 references), which allowed classification and sorting of data, so that relationships and trends in the data could be examined. The major themes were separately identified using an inductive approach [18, 26].

The data was coded into seven major themes: 'Positive aspects of TVW', 'Possible improvements to TVW', 'Alternative methods to disseminate knowledge', 'Assessment of the seizure videos', 'Control hurdles', 'Behavioral change' and 'Extra'. The key points of the seven themes were summarized with supporting statements by participants (S2 File).

## Observations

Observational data was collected at three time points per study site within the 7-week period. Pictures were taken with a smartphone and an observational checklist was filled out immediately after leaving the sites (S3 File).

## Results

The data from 47 study subjects were included in the study, with 25 and 22 from Lusaka and Katete, respectively. In Lusaka, the study participants consisted of 10 pig traders (40%) and 15 slaughter slab workers (60%; 12 slaughterers, 2 board members and 1 cashier). In Katete, no conventional pork supply chain with slaughter slabs or slaughterhouses was present. All 22 participants from Katete performed all steps from acquisition of pigs to the selling of pork and offal and will be referred to as butchers.

The study group consisted of 43 males (91.5%) and 4 females (8.5%), with all female participants being part of the Lusaka group. The ages ranged from 22 to 52 years, with an average of 36 years.

The follow-up sessions were attended by 43 of the original 47 participants (Lusaka: 24/25, Katete: 19/22). The participants with access to computers (11/47) and who consequently received USB sticks at the end of the workshop were all from the Lusaka study site.

### Questionnaires

On average, about one hour was needed to complete the questionnaires. The average baseline *T. solium* knowledge was 62% (Lusaka: 56%, Katete: 68%), with 11 participants scoring less than 50% (all from Lusaka group) and 5 participants scoring more than 75%. At baseline, 91% (43/47) of participants had heard about cysticercosis which is locally known as *masese*, *mase* or *m'sokwe*.

At the end of the workshop ('post'), the overall average score increased significantly (+20%, p<0.001) to 82% (Lusaka: 81%, Katete: 84%), with 1 participant scoring less than 50% (Lusaka group) and 36 participants scoring more than 75%. When the knowledge was tested at the '3weekspost' time point, there was no significant difference in the overall average score compared to the 'post' time point (+2.5%, p = 0.320). With an overall average score of 85% (Lusaka: 85%, Katete: 84%), the '3weekspost' knowledge remained significantly higher compared to the 'pre' time point (p<0.001) (Fig 2).

### Baseline knowledge

Averages per category ranged from 42–88% with five of the six category scores exceeding 50% (Table 1). The least understood category of *T. solium* at baseline was 'acquisition and transmission' (42%), especially the acquisition of PCC (11%) and NCC/HCC (13%). Furthermore, knowledge about PCC prevention (17%) and the relation between PCC, HCC and taeniosis (cysticercosis-taeniosis: 28%; PCC-HCC: 44%) were lacking. In total, seven individual questions received less than 50% correct answers at the 'pre' time point.

Baseline knowledge was significantly higher (p<0.001) in the Katete study group compared to the Lusaka study group; with butchers scoring significantly higher than slaughter slab workers (p<0.001) and pig traders (p = 0.007). Furthermore, participants who had owned pigs scored significantly higher (p<0.001) than those who had not (Table 2).

### Knowledge uptake

Knowledge increases from baseline were found in all categories except 'PCC in slaughtered pigs' and 'HCC/NCC', both immediately and three weeks after the use of TVW (Table 1). There were no categories with significant knowledge decreases and all six category scores exceeded 50% immediately and three weeks after the educational component.

Two individual questions received less than 50% correct answers at the 'post' and '3weekspost' time points, both concerning NCC. Firstly, the acquisition of NCC remained poorly understood ('post': 30%, '3weekspost': 33%) despite a small increase in correct answers.

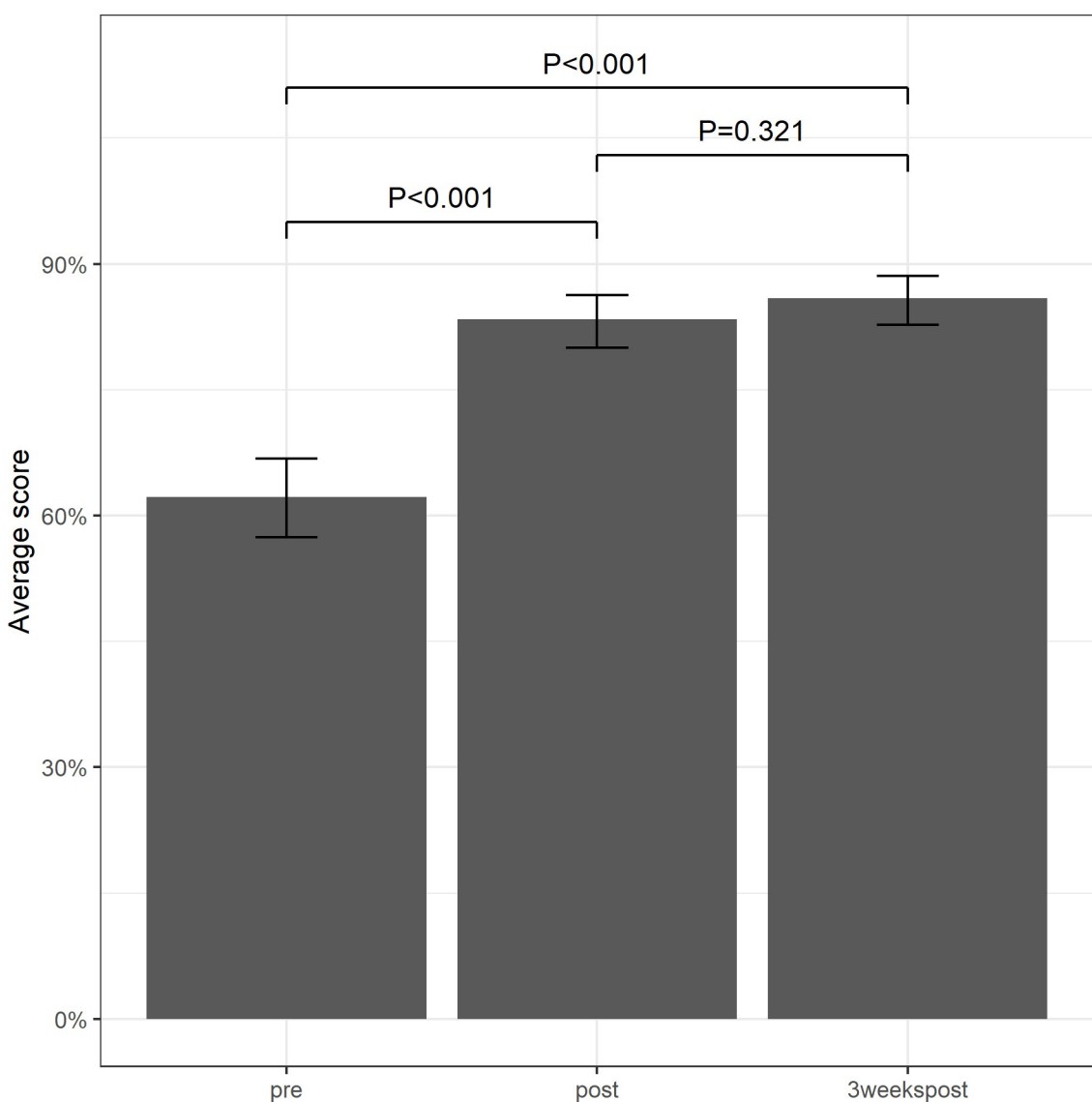

**Fig 2. Average knowledge scores over all six categories at each of the three time points.**

Secondly, the question regarding NCC's transmissibility received a lot of incorrect answers ('post': 40%%, '3weekspost': 12%) despite being relatively well answered at baseline (57%).

Knowledge uptake was significantly higher (p<0.001) in participants with a secondary school diploma than those with only a primary school diploma (Table 2). Traders had a significantly higher (p = 0.007) knowledge uptake from baseline than butchers at the end of the educational workshop.

## Perceived PCC occurrence

At the end of the educational workshop, half of the participants (23/46) indicated to detect PCC in at least one pig a week. All but one participant had observed PCC in a slaughtered pig and the perceived occurrence of PCC was higher in Katete than in Lusaka. During the short survey on the participant's general information (7 questions), all participants from Katete

**Table 1. Results from 'pre', 'post' and '3weekspost' questionnaire surveys and associated knowledge changes between the different time points.** P-values of the knowledge changes are listed between brackets.

| Category | EDUCATIONAL WORKSHOPS | | | FOLLOW-UP SESSIONS | | |
|---|---|---|---|---|---|---|
| | Pre (%) | Post-Pre | Post (%) | 3weekspost-Post | 3weeks post (%) | 3weekspost-Pre |
| *T. solium* in general | 53% | **+36% (p<0.001)** | 89% | -2.0% (p = 0.904) | 87% | **+34% (p<0.001)** |
| Acquisition & transmission | 42% | **+23% (p<0.001)** | 65% | -3.0% (p = 0.779) | 62% | **+20% (p<0.001)** |
| PCC[1] in live pigs | 53% | **+33% (p<0.001)** | 86% | +5.0% (p = 0.309) | 91% | **+38% (p<0.001)** |
| PCC[1] in slaughtered pigs | 88% | -5.0% (p = 0.398) | 83% | +13% (p = 0.003) | 96% | +8.0% (p = 0.053) |
| Taeniosis | 72% | **+20% (p<0.001)** | 92% | +5.0% (p = 0.212) | 97% | **+25% (p<0.001)** |
| HCC[2]/NCC[3] | 80% | +9.0% (p = 0.138) | 89% | +2.0% (p = 0.943) | 91% | +11% (p = 0.084) |
| **Overall** | **62%** | +20% (p<0.001) | **82%** | +3.0% (p = 0.320) | **85%** | +23% (p<0.001) |

1 PCC: porcine cysticercosis, [2]HCC: human cysticercosis, [3]NCC: neurocysticercosis

stated that meat inspection was always performed. However, 52% (13/25) of the Lusaka participants answered that meat inspection was only performed when there was a meat inspector around. Moreover, three participants from Lusaka (12%, 3/25) replied that meat inspection was never performed.

## Focus group discussions

Four FGDs were conducted and an average of 11 participants were involved in each discussion. The groups from Lusaka and Katete were referred to as L1/L2 and K1/K2, respectively. The average duration of an FGD was 30 minutes (20–40 minutes) including the demonstration of the video fragments (6 minutes in total).

**Table 2. Association between participants' characteristics and their baseline knowledge & knowledge uptake.**

| Variable | Level (n) | BASELINE KNOWLEDGE | | KNOWLEDGE UPTAKE[1] | |
|---|---|---|---|---|---|
| | | OR (95% CI) | P-value | OR (95% CI) | P-value |
| Gender | Male (43) | 1 | | 1 | |
| | Female (4) | -0.473 (-1.040; 0.094) | 0.102 | 0.020 (-0.747; 0.786) | 0.960 |
| Age (years) | age ≤ 30 (10) | 1 | | 1 | |
| | 30 < age ≤ 40 (24) | 0.341 (-0.037; -0.294) | 0.077 | -0.075 (-0.668; -1.255) | 0.803 |
| | age > 40 (11) | 0.145 (0.719; 0.585) | 0.517 | -0.601 (0.517; 0.052) | 0.071 |
| District | Katete (22) | 1 | | 1 | |
| | Lusaka (25) | **-0.531 (-0.817; -0.245)** | **<0.001** | 0.344 (-0.110; 0.799) | 0.138 |
| Job | Butcher (22) | 1 | | 1 | |
| | Slaughter slab worker[2] (12) | **-0.550 (-0.877; -0.870)** | **<0.001** | 0.068 (-0.438; 0.212) | 0.792 |
| | Trader (10) | **-0.503 (-0.224; -0.135)** | **0.007** | **0.842 (0.574; 1.471)** | **0.009** |
| Level of education | Secondary school (18) | 1 | | 1 | |
| | Primary school (26) | -0.071 (-0.404; -1.075) | 0.674 | **-0.869 (-1.372; -0.939)** | **<0.001** |
| | None (3) | -0.412 (0.261; 0.250) | 0.222 | 0.069 (-0.366; 1.078) | 0.893 |
| Pig ownership | No (27) | 1 | | 1 | |
| | Yes (20) | **0.533 (0.242; 0.824)** | **<0.001** | -0.177 (-0.640; 0.287) | 0.455 |

1 Knowledge uptake refers the knowledge change between the 'post' and 'pre' time points (educational workshop).

2 Slaughterslab workers include 12 slaughterers, 2 slaughterslab executives and 1 slaughterslab cashier.

n = number of participants

All groups generally expressed a strong positive attitude towards TVW. The most mentioned positive aspects of the program over all four groups were the educative value, the simplicity/clarity, the good illustrations, and the potential to share the knowledge.

*"I think that the program was very good because now we are protected by knowledge. Now we know how you get the tapeworm and how you get masese."* (Focus group K2)

The most mentioned potential improvements to TVW were increased accessibility (electronics needed), larger scale and addition of videos (not cartoons). The acquisition of NCC and the link with seizures was mentioned to need further clarification by both focus groups from Lusaka.

The most mentioned alternative methods to disseminate knowledge about *T. solium* were drama/sketch, radio, and booklets. In the K1 group, a participant argued that using a booklet would not be helpful because many people were illiterate. Other methods that were discussed included flyers, television, poems, and community meetings.

All groups unanimously agreed that the seizure videos were acceptable to be included in TVW, except in FGD L2 where one participant argued that "*the videos will lead to poor business as people will stop buying the pork.*" In all FGDs it was stated that the videos would help to convey the importance of *T. solium* and would promote the adoption of control measures. Participants from L2, K1 and K2 recommended to properly teach people about *T. solium* before showing the videos to avoid misunderstandings that could lead to the shunning of pork. Other mentioned benefits of the inclusion of seizure videos were improved recognition of the disease, and the tackling of the misbelief that seizures are the result of witchcraft.

The FGDs also addressed control hurdles including lack of knowledge about *T. solium*, trade of infected pigs and/or pork, free-ranging pigs, inadequate meat inspection, inadequate sanitation, consumption of undercooked pork, unknown origin of vegetables, unavailability of drugs and difficulties to maintain a good relationship with the public. Based on the FGD data, the lack of knowledge was mostly concerning the acquisition and transmission of *T. solium* infections and the relation between PCC, HCC/NCC and taeniosis. The participants referred to popular misconceptions/misbeliefs including acquisition of PCC through consumption of beer brewing residues, and seizures in humans as a result of witchcraft that had to be treated by witch doctors. A participant from the K1 group also mentioned that PCC symptoms were sometimes falsely ascribed to African Swine Fever.

*"In the villages everyone will just say that a person with epilepsy has been bewitched. Even we did not know that masese can affect people, but now we have learned."* (Focus group K1)

During the FGDs we did not explicitly ask whether the participants had adjusted their attitude or behavior after the educational workshop to prevent the provocation of socially desirable answers. Nevertheless, many participants from all groups expressed attitudinal change, and several participants from L1 mentioned behavioral change since the educational workshop. This included the examination of the tongue to detect PCC in live pigs, not selling infected pigs/pork, sending a potential NCC patient to the hospital, thoroughly cooking pork, and teaching others. The L2 group also mentioned that the USB sticks had been used by several participants during the 3-week interval.

*"I have taught at least 4 or 5 people since learning about the worm. I used what I learned, the knowledge I got from you, to teach them about masese."* (Focus group L1)

All groups reported to have observed cysticerci in pigs and seizures in both humans and pigs, with a participant from L2 stating that they had seen many people having seizures. A

participant from the K2 group also referred to the ubiquity of pig rearing, stating that almost everyone in the village had reared pigs.

## Observations

Several breaches to good hygiene practices were observed at both study sites, such as inadequate cleaning/disinfection of hands/tools/clothes, lacking protective clothing, public access to production areas, absence of a cold chain, inappropriate waste, and pest control, etc.

Observed failures in meat processing hygiene included: slaughter of a conscious pig on a patch of dirt, carcass splitting and evisceration on a soiled wet floor, piling of meat and offal from different carcasses, inappropriate carcass and meat transportation using wheelbarrows, etc. Furthermore, there was no meat inspection or official inspection observed.

## Discussion

This study indicates that educational workshops using 'The Vicious Worm' have highly significant positive effects on *T. solium* knowledge uptake and short-term retention in Zambian slaughter slab workers, butchers, and pig traders.

Baseline knowledge of *T. solium* was relatively high, especially in the Katete study group. However, the acquisition and transmission of the parasite, and the relation between PCC, HCC and taeniosis were not well comprehended; which is in line with previous studies of varying designs in Zambia and Tanzania [25, 26, 31, 50]. This lack of knowledge constitutes a major problem as it impedes the attitudinal and behavioral changes that are vital to the control of *T. solium* [25, 51, 52]. The most prominent lack of knowledge at baseline concerned the acquisition of PCC and NCC, which were also found to be the least understood aspects among Zambian primary school students when a similar questionnaire was used [31, 45].

The educational workshops using TVW significantly increased the participants' knowledge of most aspects of *T. solium*. This included considerable knowledge increases concerning PCC and taeniosis prevention, which are vital to inspire implementation of preventive measures. However, the acquisition and transmissibility of HCC/NCC remained poorly understood after the use of TVW, which was also described by Hobbs et al. and Ertel et al. [26, 31]. Moreover, the decrease in correct answers regarding the transmissibility of NCC after use of TVW was also found in Zambian primary school children [31]. Insufficient understanding of the *T. solium* lifecycle despite health education has been reported after diverse educational interventions among different target groups [22–24, 26, 27, 29, 31]. The most common misconceptions after the educational session were the acquisition of NCC through consumption of undercooked pork and the transmission of NCC through stool. This likely reflects the complexity of the parasite's life cycle and calls for the clarification or simplification of the NCC aspect in future versions of the software. This notion is further strengthened by the FGDs in which two participants proposed to emphasize the NCC aspects during TVW health education.

Our study found a better baseline knowledge in pig owners, butchers, and participants from Katete; however, the latter two groups overlapped. The increase in knowledge was more prominent in participants with a secondary school diploma than those with only a primary school diploma. This suggests that further simplifications of TVW might be beneficial to ensure a similar knowledge uptake among people with less advanced educational levels. Furthermore, a bigger knowledge increase was found in pig traders than in butchers, with no significant difference between butchers and slaughter slab workers. The small number of female participants (4) and participants without a diploma (3) impeded their comparison with other groups.

The FGD data confirmed the lack of knowledge regarding the acquisition and transmission of the parasite and the relation between PCC, HCC/NCC and taeniosis in the communities.

Furthermore, the FGDs exposed the persistence of certain misbeliefs concerning PCC and NCC. Locally porcine cysticercosis is called with the terms *masese*, *m'sokwe* and *mase*, which literally translates to *'beer dreg'* due to the resemblance of beer brewing residues with white nodular cysticerci. Therefore, a popular misbelief suggested that pigs acquire PCC after the consumption of beer brewing residues, which pigs were sometimes fed. Another popular misbelief alleged that epilepsy patients were bewitched and could be healed by witch doctors. A 2010 study by Thys et al. found the same misbeliefs to be popular in communities from the Petauke district of the Eastern province of Zambia [17]. These misperceptions highlight the potential of health education in endemic areas, where communities are unable to take well-advised precautionary measures.

The study participants expressed a positive attitude towards TVW, describing the program as simple, clear, and educative. A similar positive attitude towards the program was observed amongst Tanzanian medical and agricultural professionals [26]. The FGD data also suggest that some of the participants who received a USB stick with the software voluntarily used the software between the initial and follow-up visits. Moreover, the participants appeared committed to share the knowledge they had acquired from TVW, with several participants stating to have done so during the three weeks after the initial visit. Previous studies also reported knowledge transfer from educated individuals to the community [23, 28, 30] but more research is needed to assess the efficacy and effectiveness of this information transfer.

During the FGDs, participants proposed potential improvements, including support for more languages, and the addition of videos. Videos depicting seizures in pigs and humans could also enhance the software as it could help to convey the importance of *T. solium* and promote the adoption of control measures. During the FGDs, the accessibility was indicated as a limitation of the tool as it requires a computer, smartphone, or tablet. Therefore, the organization of group workshops with a moderator might boost the dissemination of TVW's information; these are quite easy to organize and only require a beamer, a computer, and a small generator in the absence of electrical supply.

The observed breaches to good hygiene practices and meat processing hygiene demonstrated the importance of the pork supply chain workers in the propagation of foodborne parasites like *T. solium*. Furthermore, the observed lack of meat inspection or any type of official control passes the responsibility for distribution and consumption of infected pork to non-trained individuals. Therefore, slaughter slab workers, pig traders, butchers and meat inspectors could serve as target groups for future educational interventions. In addition to tackling the lack of knowledge, it could lead to long-term behavioral changes that improve PCC detection and reduce the distribution of infected pork and pigs. Furthermore, these pork supply chain workers were in close contact with pig farmers and pork consumers with whom they could share knowledge, possibly inspiring the adoption of control measures. However, increased knowledge does not automatically translate into attitudinal and behavioral change and economic and/or socio-cultural factors can override the perceived and rather long-term health risk [18, 25, 53]. Nevertheless, our preliminary FGD data suggest incipient attitudinal and behavioral changes.

The study would have benefited from a larger number of study subjects and a control group. Social desirability bias and unanswered questions (mainly 'pre' questionnaire) could also affect the validity and reliability of the results. Another limitation of the study is the reliance on written English for the questionnaires, which could potentially have contributed to misunderstandings. Assessments of long-term effects, the extent of knowledge transfer and behavioral change associated with the use of TVW were outside the scope of this study but would be useful in future studies.

The results demonstrated that TVW is an effective short-term health education tool for key actors in the Zambian pork supply chain. Translation into other languages and adaptations to

different sociocultural and regional contexts could allow global implementation of the specific health education tool. However, the software could benefit from the addition of videos and more emphasis on the NCC aspect of TSTC. Moreover, we recommend a well-thought-out and unhurried teaching approach in future group workshops to prevent confusion. In conclusion, the highly significant effect on knowledge uptake and positive attitude towards the program call for consideration of TVW in future intervention programs.

## Supporting information

**S1 Data. Questionnaire scores per participant and per question.** Spreadsheet with assessment of the answers per question, and per category.
(XLSX)

**S1 File. The Vicious Worm Question List.** A list of questions asked during the 'pre', 'post' and '3weekspost' questionnaire survey.
(PDF)

**S2 File. Key points FGDs.** Overview of key points that were discussed during the focus group discussions.
(PDF)

**S3 File. Observation checklist.** The checklist used to collect observational data.
(PDF)

## Acknowledgments

The authors are grateful for the enthusiastic participation and assistance of the pork supply chain workers in Lusaka and Katete. We would also like to thank the Chibolya Small Livestock Cooperative, and the Katete veterinary assistant Justine Mwango for their help in the selection of study participants.

## Author Contributions

**Conceptualization:** Victor Vaernewyck, Sarah Gabriël, Chiara Trevisan.

**Data curation:** Victor Vaernewyck, Kabemba Evans Mwape, Chishimba Mubanga, Brecht Devleesschauwer, Chiara Trevisan.

**Formal analysis:** Brecht Devleesschauwer.

**Funding acquisition:** Sarah Gabriël.

**Investigation:** Victor Vaernewyck, Kabemba Evans Mwape, Chishimba Mubanga, Sarah Gabriël, Chiara Trevisan.

**Methodology:** Victor Vaernewyck, Kabemba Evans Mwape, Brecht Devleesschauwer, Sarah Gabriël, Chiara Trevisan.

**Project administration:** Sarah Gabriël, Chiara Trevisan.

**Supervision:** Kabemba Evans Mwape, Sarah Gabriël, Chiara Trevisan.

**Visualization:** Victor Vaernewyck.

**Writing – original draft:** Victor Vaernewyck.

**Writing – review & editing:** Victor Vaernewyck, Kabemba Evans Mwape, Chishimba Mubanga, Brecht Devleesschauwer, Sarah Gabriël, Chiara Trevisan.

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
