## [Decision Letter · Decision Letter 0]

17 Aug 2020

Dear Dr. Trevisan,

Thank you very much for submitting your manuscript "Effects of ‘The Vicious Worm’ educational software on Taenia solium knowledge among key pork supply chain workers in Zambia" for consideration at PLOS Neglected Tropical Diseases. As with all papers reviewed by the journal, your manuscript was reviewed by members of the editorial board and by several independent reviewers. The reviewers appreciated the attention to an important topic. Based on the reviews, we are likely to accept this manuscript for publication, providing that you modify the manuscript according to the review recommendations. 

Sincerely,

Keke Fairfax, PhD

Deputy Editor

Keke Fairfax

Deputy Editor

Reviewer's Responses to Questions

**Key Review Criteria Required for Acceptance?**

**Methods**

-Are the objectives of the study clearly articulated with a clear testable hypothesis stated?

-Is the study design appropriate to address the stated objectives?

-Is the population clearly described and appropriate for the hypothesis being tested?

-Is the sample size sufficient to ensure adequate power to address the hypothesis being tested?

-Were correct statistical analysis used to support conclusions?

-Are there concerns about ethical or regulatory requirements being met?

Reviewer #1: The objectives are clear. The study design only allows the short term evaluation, which is mentioned in the manuscript but not in the abstract conclusions. Population is clearly described. I am not in a position to comment on the statistics. No concerns about ethical or regulatory requirements.

Reviewer #2: One major concern I have with the manuscript is the reliance on written English (rather than written Nyanja-Chewa) for the questionnaires. This potentially contributes to misunderstandings. Indeed, the authors themselves point out an example of potential language/translation issues: “A popular misbelief suggested that pigs acquire PCC after the consumption of beer brewing residues, which pigs were sometimes fed. This might be due to the use of the local terms masese, m’sokwe and mase to describe the white nodular cysticerci while these terms literally translate to ‘beer dregs’.” This made me less enthusiastic about the study's results.

Line 260: How much was the financial incentive for study participation?

**Results**

-Does the analysis presented match the analysis plan?

-Are the results clearly and completely presented?

-Are the figures (Tables, Images) of sufficient quality for clarity?

Reviewer #1: Yes.

Reviewer #2: Abstract:

Background section is disjointed

What is a “slaughterslab” worker? It would be helpful if the authors could expand upon what pig-related activities are performed at a slaughterslab (presumably killing the pigs, but any other activities?).

**Conclusions**

-Are the conclusions supported by the data presented?

-Are the limitations of analysis clearly described?

-Do the authors discuss how these data can be helpful to advance our understanding of the topic under study?

-Is public health relevance addressed?

Reviewer #1: The conclusions in the abstract over-interprets the value of the results. It concludes that the results shows TVW to be effective... and should be considered for integration in T. solium control programs. The conclusion in the abstract needs to clarify that it shows to be effective in the short term, and could (instead of should) be considered for integration. 

The limitations are reasonably presented.

Reviewer #2: Yes

**Editorial and Data Presentation Modifications?**

Reviewer #1: (No Response)

Reviewer #2: (No Response)

**Summary and General Comments**

Reviewer #1: This paper is about the short-term effects of “The Vicious Worm” (TVW) educational software on Taenia solium knowledge among key pork supply chain workers in Zambia. The authors showed an improved knowledge 3 weeks after workshops with the software using a questionnaire and FDG. 

The manuscript is in general well written and clear. The main issue with the manuscript is that it over-interprets the value of the results in the abstract when it concludes that the results shows TVW to be effective... and should be considered for integration in T. solium control programs. The conclusion needs to clarify that it shows to be effective in the short term, and could be considered for integration. 

Abstract:

1. Line 34: Health education is regarded important but not essential. Suggest change the word essential to important. 

2. Lines 52-55: The conclusions seem to over-interpret the value of “The Vicious Worm”. Indeed, it seems that is a useful and promising tool, but it was evaluated only 3 weeks after its use, so it is too soon to evaluate its impact and recommend its integration. Suggest toning down the conclusion/significance as mentioned above. 

Author summary:

3. Line 60: The word chemotherapy needs to be reviewed. If it refers to humans, it should be Preventive chemotherapy (chemotherapy alone is usually referred to cancer treatment – preventive chemotherapy refers to treating populations at risk). If the authors refer to pigs, the treatment with oxfendazole (or similar interventions in animals) is classified as metaphylaxis or medication, not as chemotherapy. To avoid discussion and confusion, if the authors refer to pigs, I suggest using the word medication.

4. Line 60: add pigs after vaccination, to clarify it refers to the vaccination of pigs and not people. 

Manuscript:

5. Line 96: What do you mean with pig farming as a transitory activity? 

6. Was any incentive given to the participants in the questionnaires or FDG? 

7. Line 451: note that behavioural change in references 28 and 30 was in different groups of people, with different educational background which has an impact as mentioned in line 404. The authors imply that what they found in pork supply chain workers would/could be as described in references 28 and 30, however the very substantial differences in the subjects between the different studies does not permit this conclusion. Ref 28 relates to vets, extension officers, medical officers, etc (Ertel) while in this paper are pig traders, slab workers and butchers which have a different education level. Ref 30 refers to school children. 

8. Line 458: needs to clarify it is in the short term

Reviewer #2: (No Response)

PLOS authors have the option to publish the peer review history of their article (what does this mean?). If published, this will include your full peer review and any attached files.

Reviewer #1: No

Reviewer #2: No
---

## [Editor Report · Decision Letter 1]

10 Sep 2020

Dear Dr. Trevisan,

We are pleased to inform you that your manuscript 'Effects of ‘The Vicious Worm’ educational software on Taenia solium knowledge among key pork supply chain workers in Zambia' has been provisionally accepted for publication in PLOS Neglected Tropical Diseases.

Best regards,

Keke Fairfax, PhD

Deputy Editor

Keke Fairfax

Deputy Editor

---

## [Editor Report · Acceptance letter]

12 Oct 2020

Dear Dr. Trevisan,

We are delighted to inform you that your manuscript, "Effects of ‘The Vicious Worm’ educational software on *Taenia solium* knowledge among key pork supply chain workers in Zambia," has been formally accepted for publication in PLOS Neglected Tropical Diseases.

Best regards,

Shaden Kamhawi

co-Editor-in-Chief

Paul Brindley

co-Editor-in-Chief
